

# Exact acceleration of complex real-time model checking based on overlapping cycle

Guoqing Wang[1], Lei Zhuang[1], Yu Song[1], Mengyang He[1], Ding Ma[2] and Ling Ma[1,3]

[1] School of Information Engineering, Zhengzhou University, Zhengzhou, Henan, China
[2] College of Information Science and Engineering, Henan University of Technology, Zhengzhou, Henan, China
[3] Digital Medical Image Technique Research Center, Zhengzhou University, Zhengzhou, Henan, China

## ABSTRACT

When real-time systems are modeled as timed automata, different time scales may lead to substantial fragmentation of the symbolic state space. Exact acceleration solves the fragmentation problem without changing system reachability. The relatively mature technology of exact acceleration has been used with an appended cycle or a parking cycle, which can be applied to the calculation of a single acceleratable cycle model. Using these two technologies to develop a complex real-time model requires additional states and consumes a large amount of time cost, thereby influencing acceleration efficiency. In this paper, a complex real-time exact acceleration method based on an overlapping cycle is proposed, which is an application scenario extension of the parking-cycle technique. By comprehensively analyzing the accelerating impacts of multiple acceleratable cycles, it is only necessary to add a single overlapping period with a fixed length without relying on the windows of acceleratable cycles. Experimental results show that the proposed timed automaton model is simple and effectively decreases the time costs of exact acceleration. For the complex real-time system model, the method based on an overlapping cycle can accelerate the large scale and concurrent states which cannot be solved by the original exact acceleration theory.

## INTRODUCTION

In real-time embedded systems, especially complex real-time control systems, discrete logic control and continuous time behavior depend on and influence each other. Take the Internet of things (IoT) gateway security system (*Wang et al., 2018*) as an example: its control center generally has many different control modes to deal with diverse security risks, such as tampering, intrusion, and identity forging. Important system parameters (e.g., sensor status, monitoring instructions, and terminal feedback information) change continuously over time. To meet specific time constraints or parameter values in the IoT gateway security system, the management mode must be adjusted over time. The change rules of important parameters also differ by mode, and the response time to

Corresponding author
Lei Zhuang, ielzhuang@zzu.edu.cn

various events should be modified accordingly. In this type of system (*Lee et al., 2019*), logic control describes the logical control transformation of the system through models with high abstraction levels, such as finite state machine and Petri net. Time behavior can be simulated by clock variables and clock zone transformation. Between the two layers, signals of the continuous layer and control modes of the discrete layer are correlated and transformed by certain interfaces and rules.

Typically, test and simulation technologies are the main means of guaranteeing software quality; however, they cover problems when using the operating system as the main measure, which cannot guarantee test completeness. These approaches are thus incapable of traversing all states in a real-time system, leading to covert problems in system operations (*Wang, Pastore & Briand, 2019*). In the field of security-related systems with zero tolerance for system error, using formal theory and technology for security authentication results in clear descriptions and avoids the complexity of safety verification. Formal description analysis and refinement have thus become a focus of recent research in related fields.

In real-time model checking, timed automata can model the temporal behavior of real-time systems (*Pinisetty et al., 2017*). Clocks describe the state transitions, and clock constraints serve as the theoretical basis for real-time system model checking (*Han, Yang & Xing, 2015*). This approach can easily realize automatic combination and transformation with other methods. The method is widely used in polling control systems, railway interlocking systems, and similar applications. Due to clock variables, control programs and external environments often use different time measures, which can cause the number of states to increase exponentially when a timed automaton is transformed into a zone automaton. The reachability analysis algorithm generates many state fragments (*Iversen et al., 2000*; *Chen & Cui, 2016*), resulting in a sharp increase in the state space and considerably prolonged detection time.

The acceleration technique is a reduction method used to solve the fragmentation problem following from time measurement differences. *Dubout & Fleuret (2013)* applied an acceleration technique to linear target detection and effectively improved the detection performance. *Jeong et al. (2014)* applied an implicit Markov model as an improved framework to accelerate the inference model. For distributed and parallel computing, a workstation and a multicore processor were used to accelerate state-space searching (*Konur, Fisher & Schewe, 2013*). *Lin, Chen & Xu (2017)* studied an acceleration model using a Bayesian classifier by analyzing the behavior of heterogeneous population trends; results indicated that acceleration in the reliability assessment improved the analytic accuracy. The model checking of linear temporal logic (LTL) model was studied by *Barnat et al. (2010)*, which employed computed unified device architecture for acceleration. Two SAT problem solvers were used to validate online models and accelerate the processing of complex behaviors (*Qanadilo, Samara & Zhao, 2013*).

The reachability problem is the first to consider in timed automata, which determines whether a path exists from its initial state to a target state. This problem can be solved by computing the zones that apply the abstraction technique in practice. State-of-the-art abstraction methods (*Behrmann et al., 2006*; *Herbreteau, Srivathsan & Walukiewicz, 2016*) produce an approximation closer to the actual reachable clock valuation, which includes

coarser abstractions. Exact acceleration is an excellent means of abstraction to reduce required storage space and can alleviate state-space explosion. For practical issues such as protocol validation (*Zhang et al., 2013*), IoT system modeling (*Li et al., 2013*), and smart contract security verification in blockchain (*Cruz, Kaji & Yanai, 2018*; *Grishchenko, Maffei & Schneidewind, 2018*), exact acceleration technology is an efficient way of minimizing required storage space and time.

When *Iversen et al. (2000)* used UPPAAL to verify the LEGO robotic system, a fragmentation problem was identified and briefly described, and some ideas for further research were suggested. An approximation technique was applied to a real-time system model for security and connectivity analysis, which avoided repetitive control (*Möller, 2002*). After that, a real-time property language $L_{\forall S}$ was proposed to check the rejection state of reachability and reduce safety and boundary liveness simultaneously (*Aceto et al., 2003*). The problems and methods in these publications have promoted the concept of exact acceleration and inspired further research. Related studies on exact acceleration in real-time model checking include *Hendriks & Larsen (2002)*, *Yin, Song & Zhuang (2010)*, *Yin, Zhuang & Wang (2011)*, *Gou et al. (2014)*, *Boudjadar et al. (2016)*, and *Chadli et al. (2018)*. In the following four examples, the window of the acceleratable cycle is $[a, b]$.

- *Hendriks & Larsen (2002)* introduced a method of syntax adjustment to a subset of timed automata by adding an appended cycle whose length was $\lceil a/(b-a) \rceil$ times longer than that of the acceleratable cycle. This method accelerates forward symbolic reachability analysis, which solves the fragmentation problem and optimizes the verification of the LEGO robotic system.
- *Yin, Song & Zhuang (2010)* proposed a method to identify the acceleratable cycle in timed automata by introducing topological sorting for a large state space of a timed automaton; by simplifying the scale of timed automata, the method operated efficiently.
- An exact acceleration method based on a parking cycle was proposed (*Yin, Zhuang & Wang, 2011*), in which the entry boundary condition was determined by the size of the acceleratable cycle's window (the condition is $z \geq a \times \frac{a}{b-a} + n_0$); the automaton model improved the speed of exact acceleration and reduced the cost.
- By analyzing the main parameters of the acceleration process, *Gou et al. (2014)* proposed a method for determining whether exact acceleration was required. This approach can be used to avoid adding an appended cycle to reduce verification speed when the number of fragments is small, or fragments do not satisfy certain conditions.
- *Boudjadar et al. (2016)* proposed a development method to improve the utilization rate of resources by using model-checking technology. In the design and development stage, exact acceleration technology was used to greatly improve the capability of symbolic model checking in a processing scheduling system. For the scheduling problem of network physical systems, *Chadli et al. (2018)* modeled advanced specifications and validation frameworks with the help of exact acceleration technology, automatically transforming high-level specifications into formal models. The above two research works mainly applied exact acceleration to model a system resource scheduling problem but did not improve the original exact acceleration theory.

When modeling a complex real-time system (*Wang et al., 2019*), multiple acceleratable cycles may overlap at the same location. If the appended cycle method is used for exact acceleration, then the added locations multiply as the number of acceleratable cycles increases, resulting in insufficient memory for model checking. If the parking cycle method is used for exact acceleration, acceleratable-cycle stacking leads to non-uniformity in parking-cycle entry conditions; differences in the windows of multiple acceleratable cycles can increase time consumption drastically. In this paper, we propose an exact acceleration method for complex real-time model checking based on an overlapping cycle, which is an application scenario extension of parking-cycle technique. A single overlapping cycle is developed by comprehensively analyzing the accelerating effects of multiple acceleratable cycles and analyzing acceleration differences among these cycles. The overlapping cycle is simple to create and has a fixed length, eliminating the need to add multiple locations for complex real-time models. The overlapping cycle adds much less state space than appended cycles or parking cycles in model checking, substantially reducing the acceleration cost. The proposed method can be effectively applied to modeling and verification of complex real-time systems such as the IoT gateway security system. It can also alleviate additional consumption of time and space caused by state-space explosion while maintaining the original nature of the system.

The remainder of this paper is organized as follows. The section 'Preliminaries' briefly introduces timed automata, forward symbolic reachability analysis, and the theory of exact acceleration. The exact acceleration method for complex real-time models based on an overlapping cycle is proposed in 'Exact Acceleration of Complex Real-time System Model Based on Overlapping Cycle', which outlines the method of creating a single, fixed-length overlapping cycle. A timed automaton with an overlapping cycle is shown to accelerate the originally timed automaton with reachability. In 'Experimental Results', the acceleration effects of the appended cycle, parking cycle, and overlapping cycle with a complex real-time model example are compared using experiments. Finally, the 'Conclusion' provides a few ideas for future research.

## PRELIMINARIES

### Timed automata

This part is based on work by *Alur & Dill (1994)*. To illustrate the real-time clock of timed automata more clearly, we define a clock constraint set $T(C)$ contain all clock constraints. We assume that the set of clock variables is $C$, and the definition of the set of clock constraints $\tau$ is as follows:

$$\tau := c \sim n | \tau_1 \wedge \tau_2$$

where $c \in C$, $n \in \mathbb{N}$, and $\sim$ denotes one of the binary relationships $\{<, \leq, =, \geq, >\}$. The clock constraint set $T(C)$ is the set of all clock constraints $\tau$.

A clock interpretation $v$ is a mapping from $C$ to $\mathbb{R}^+ \cup \{0\}$, where $\mathbb{R}^+$ represents the set of positive real numbers. Note that $v$ assigns each clock variable in the set of clock variables $C$. For a set $X \subseteq C$, $X := 0$ indicates that $X$ assigns 0 to each $c \in X$ (i.e., clock reset), whereas the clock variables in set $C - X$ have no effects.

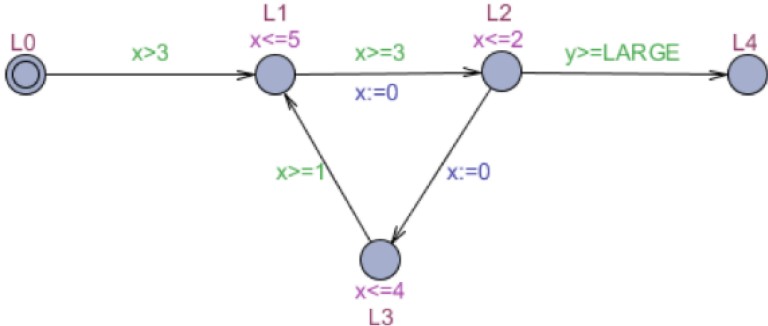

**Figure 1** Timed automaton *M*.

**Definition 1** (Timed automaton). A timed automaton is defined as a six-tuple $(C, L, L_0, A, I, E)$, where $C$ is a set of clocks, $L$ is a finite set of locations, $L_0 \subseteq L$ is the set of initial locations, $A$ is a set of action events, $I$ represents mapping that provides every location $l \in L$ with some clock constraint in $T(C)$, and $E \subseteq L \times A \times T(C) \times 2^C \times L$ is a set of edges. An edge $(l, a, \tau, \lambda, l')$ denotes a transition: when the clock constraint in location $l$ satisfies $\tau$, the system can complete action event $a$, move from location $l$ to location $l'$, and allow clocks in $\lambda$ to be reset.

Figure 1 shows an example of a timed automaton. The timed automaton $M$ represents a plain and abstract model of the control program and the external environment in a real-time system. If the control program sends instructions to the control center in an IoT security system, the environment will be decided by sensors and actuators. The cycle of locations L1, L2, and L3 model the control program labeled the control cycle, consisting of three atomic instructions, whose clock is $x$. The external environment is modeled by clock $y$, which is checked each time in L2. The clock $y$ also called global clock. The size of the threshold constant LARGE determines how slow the environment is relative to the control program. If $y \geq$ LARGE, the control cycle may be exited.

The semantics of a timed automaton $M$ is defined by a transition system $S(M)$ with *Alur & Dill (1994)*. A state of $S(M)$ is a pair $(l, v)$, where $l$ is a location of $M$ and $v$ indicates a clock interpretation for $C$ such that $v$ satisfies $I(l)$. Regarding this transition system, the traces of a timed automaton have been defined by *Hendriks & Larsen (2002)*.

## Forward symbolic reachability analysis

The forward symbolic reachability analysis algorithm is a core of the real-time model-checking tool UPPAAL (*Behrmann, David & Larsen, 2004*). The model-checking engine uses an on-the-fly strategy to search forward from the initial location to determine whether a symbolic state is reachable. For each symbolic state that has not yet been explored, it is necessary to calculate subsequent states based on their clocks and actions and compare them to searched symbolic states. If they have been seen in the past, they are discarded; otherwise, they are added to the list of explored symbolic states.

The reachability property $\varphi$ of a timed automaton $M$ can be presented as the timed computation tree logic (TCTL) formula $E <> (P)$, where $P$ is a state property of $M$.

**Table 1  Results of symbolic states from a forward symbolic exploration by timed automaton $M$.**

| State | Location | | Zone | |
|---|---|---|---|---|
| 1 | L0 | $y = 0$ | $x = 0$ | $y - x = 0$ |
| 2 | L1 | $3 < y \leq 5$ | $3 < x \leq 5$ | $y - x = 0$ |
| 3 | L2 | $3 < y \leq 7$ | $0 \leq x \leq 2$ | $3 < y - x \leq 5$ |
| 4 | L3 | $3 < y \leq 11$ | $0 \leq x \leq 4$ | $3 < y - x \leq 7$ |
| 5 | L1 | $4 < y \leq 12$ | $1 \leq x \leq 5$ | $3 < y - x \leq 7$ |
| 6 | L2 | $6 < y \leq 14$ | $0 \leq x \leq 2$ | $6 < y - x \leq 12$ |
| 7 | L3 | $6 < y \leq 18$ | $0 \leq x \leq 4$ | $6 < y - x \leq 14$ |
| 8 | L1 | $7 < y \leq 19$ | $1 \leq x \leq 5$ | $6 < y - x \leq 14$ |
| 9 | L2 | $9 < y \leq 21$ | $0 \leq x \leq 2$ | $9 < y - x \leq 19$ |
| 10 | L3 | $9 < y \leq 25$ | $0 \leq x \leq 4$ | $9 < y - x \leq 21$ |
| 11 | L1 | $10 < y \leq 26$ | $1 \leq x \leq 5$ | $9 < y - x \leq 21$ |
| 12 | L2 | $12 < y \leq 28$ | $0 \leq x \leq 2$ | $12 < y - x \leq 26$ |

We describe that $M$ satisfies $\varphi$, denoted by $M \vDash \varphi$, if a trace exists in the form of $((l_0, v_0), (l_1, v_1), \ldots) \in Tr(M)$, where $(l_i, v_i) \vDash P$ for some $i \geq 0$.

To describe the process of forward symbolic reachability analysis, we take automaton $M$ in Fig. 1 as an example. Table 1 shows the symbolic states that timed automaton $M$ searches forward from the initial location after one execution.

In Table 1, symbolic states 6 and 3 are both L2. However, the clock zones are not identical; these states represent two different symbolic states to be further forward searched. Therefore, every execution of the control cycle results in new symbolic states. Because the threshold LARGE is usually larger and clock $y$ is especially smaller, the timed automaton $M$ must execute a certain number of control cycles to increase clock $y$ effectively when verifying the reachability of L4. The number of executions depends on LARGE; if LARGE is large, then there are more executions. Due to different clocks, when the model checking tools detect a symbolic model cycle by cycle, many unnecessary clock fragments may appear in the state space, causing a forward symbolic fragment problem. For example, if we only observe the symbolic states 3, 6, 9 and 12 of location L2, we can find that each clock zone overlaps with the zone in front of it, which is called clock zone continuous. At this time, because of the small-time measurement in the cycle, the overlapped clock zone is divided into infinite segments, which leads to the fragmentation problem. Table 1 lists results from the UPPAAL simulator.

## Exact acceleration

*Hendriks & Larsen (2002)* proposed the concept of exact acceleration, based on which, we provide basic definitions for our study. The acceleratable cycle is a key concept in exact acceleration. An acceleratable cycle can use only one clock in clock constraints (including invariants, guards, and resets).

**Definition 2** (Acceleratable cycle). Let $M = (C, L, l_0, A, I, E)$ be a timed automaton, $E_c = (e_0, \ldots, e_{n-1}) \in E^n$, and $x \in C$. An acceleratable cycle is defined as a two-tuple $(E_c, x)$ when the following conditions are satisfied:

- $E_c$ is a cycle;
- for all locations in $E_c$, $I(l)$ is either empty or in the form of $\{x \leq c\}$;
- if $(l, a, \tau, \lambda, l') \in E_c$, then $\tau$ is empty or in the form of $\{x \geq c\}$, and $\lambda$ is empty or only contains $x$; and
- $x$ must be reset on all in-going edges to $src(e_0)$.

Clock $x$ is the clock of the cycle, $I(l)$ is the location invariant, and $\tau$ is the edge guard. The location $src(e_0)$ is the reset location whose out-going edge is $e_0$, which indicates the external clock's checking position in the acceleratable cycle. If a specific location's in-going edge is $e_i$ in the cycle, then the out-going edge of this position is $e_{i+1}$, where $i \in [0, n-2]$.

The cycle in automaton $M$ (Fig. 1), composed of locations L1, L2, and L3, is an acceleratable cycle. The clock of the cycle is $x$, and the reset location is L2. The invariants and guards are in accordance with the defined form of clock $x$, and the clock resets to zero at the only in-going edge of L2.

**Definition 3** (Window of acceleratable cycle). Let an acceleratable cycle in the timed automaton $M$ be $(E_c, x)$. The compression sequence of all traces is expressed as $T_r(E_c) = \left( (l_0, v_0), \left( l_0, v_0' \right), (l_1, v_1), \ldots, (l_{n-1}, v_{n-1}), \left( l_{n-1}, v_{n-1}' \right), (l_n, v_n) \right)$, where $v_i$ and $v_i'$ indicate different clock interpretations, $i \in [0, n]$, and $l_0 = l_n = src(e_0)$. $((l_j, v_j'), (l_{j+1}, v_{j+1}))$ depends on the edge $e_j$ and can be understood as an action event of $e_j$, $j \in [0, n-1]$. The window of the acceleratable cycle $(E_c, x)$ is defined as the interval $[a, b]$, $a, b \in \mathbb{N}$ when the following conditions are satisfied:

- the total delay of $T_r(E_c)$ is an element of $[a, b]$; and
- for any real number $d \in [a, b]$, we adjust the delays under legal clock constraints in $T_r(E_c)$ to ensure the total delay is $d$.

The meaning of the total delay in this definition is an increase in the clock of the cycle, which can be simply defined as the increment of the external clock when the acceleratable cycle returns to the reset location once from the initial location. The window is the minimal and maximal time it may take to pass through a cycle. According to this definition, the window of the acceleratable cycle shown in Fig. 1 can be calculated as $[3, 7]$.

**Definition 4** (Accelerated automaton based on appended cycle). Let $M = (C, L, l_0, A, I, E)$ be a timed automaton, and let $Cycle = (E_c, x)$ be an acceleratable cycle of $M$, where $L = \{l_0, l_1, \ldots, l_m\}$, $E_c = (e_0, e_1, \ldots, e_{n-1})$, $e_i = (l_i, a_i, \tau_i, \lambda_i, l_{i+1})$. Acceleration of $M$ based on the appended cycle is a new automaton $Acc_a(M, Cycle)$ defined as $(C, L', l_0, A, I', E')$, where

- $L' = L \cup \left\{ l_1', l_2', \ldots, l_{n-1}' \right\} \cup \left\{ l_0' \right\} \cup \left\{ l_1'', l_2'', \ldots, l_{n-1}'' \right\}$
- $I'(l_i) = I(l_i)$, $0 \leq i \leq m$
- $I'\left( l_i' \right) = I(l_i)$, $1 \leq i \leq n-1$
- $I'\left( l_0' \right) = \varnothing$
- $I'\left( l_i'' \right) = I(l_i)$, $1 \leq i \leq n-1$

- $E' = E \cup \{(l_0, a_0, \tau_0, \lambda_0, l'_1), (l'_{n-1}, a_{n-1}, \tau_{n-1}, \lambda_{n-1}, l'_0)\}$
  $\cup \{(l'_0, a_0, \tau_0, \lambda_0, l''_1), (l''_{n-1}, a_{n-1}, \tau_{n-1}, \lambda_{n-1}, l_0)\}$
  $\cup \{(l'_i, a_i, \tau_i, \lambda_i, l'_{i+1}), (l''_i, a_i, \tau_i, \lambda_i, l''_{i+1}) | 1 \le i \le n-1\}$
- in particular, $E' = E \cup \{(l_0, a_0, \tau_0, \lambda_0, l'_0), (l'_0, a_0, \tau_0, \lambda_0, l_0)\}$ when $n = 1$.

**Theorem 1**. Let $M = (C, L, l_0, A, I, E)$ be a timed automaton, and let $Cycle = (E_c, x)$ be an acceleratable cycle of $M$ with a window $[a, b]$. If $\varphi$ is the reachability property of $M$, then

$$3a \le 2b \Rightarrow (M \vDash \varphi \Leftrightarrow Acc_a(M, Cycle) \vDash \varphi).$$

Theorem 1 has been proved in *Hendriks & Larsen (2002)*. In Definition 4, the appended cycle is obtained by expanding the acceleratable cycle twice. If the timed automaton $M$ is added to the appended cycle by expanding the acceleratable cycle $i$ times, then the precondition in Theorem 1 can be generalized to $(i+1)a \le ib$.

**Definition 5** (Accelerated automaton based on parking cycle). Let $M = (C, L, l_0, A, I, E)$ be a timed automaton, and let $Cycle = (E_c, x)$ be an acceleratable cycle of $M$ with a window of $[a, b]$, where $L = \{l_0, l_1, \ldots, l_m\}$, $E_c = (e_0, e_1, \ldots, e_{n-1})$, $e_i = (l_i, a_i, \tau_i, \lambda_i, l_{i+1})$. The global clock is $y$, and the maximum value of $y$ before entering the acceleratable cycle is $n_0$. The acceleration of $M$ based on the parking cycle is a new automaton $Acc_p(M, Cycle)$ defined as $(C, L', l_0, A, I', E')$, where

- $L' = L \cup \{l'_0\}$
- $I'(l_i) = I(l_i)$, $0 \le i \le m$
- $I'(l'_0) = \varnothing$
- $E' = E \cup \{(l_0, a_0, \tau', \varnothing, l'_0), (l'_0, a_{n-1}, \varnothing, \lambda_{n-1}, l_0)\}$, $\tau'$ is $y \ge a \times \lceil \frac{a}{b-a} \rceil + n_0$.

Definition 5 has been given in *Yin, Zhuang & Wang (2011)*. The accelerated automaton $Acc_a(M, Cycle)$ equals the timed automaton $M$ with an appended cycle composed of locations $l_0, l'_1, l'_2, \ldots, l'_{n-1}, l'_0, l''_1, l''_2, \ldots, l''_{n-1}$. Accelerated automaton $Acc_p(M, Cycle)$ equals the timed automaton $M$ with a parking cycle whose edge guard $y$ controls the acceleration timing. Only when the acceleratable cycle has been executed at least $\lceil a/(b-a) \rceil$ times is the automaton permitted to enter the parking cycle. Figures 2 and 3 display the acceleration of $M$ (Fig. 1) based on the appended cycle and parking cycle, respectively. They are labeled the accelerated automata $M_a$ and $M_p$. Because the window of the acceleratable cycle is $[3, 7]$, the edge guard of the parking cycle is $y \ge 3$ in $M_p$.

**Theorem 2**. Let $M = (C, L, l_0, A, I, E)$ be a timed automaton, and let $Cycle = (E_c, x)$ be an acceleratable cycle of $M$ with a window of $[a, b]$. If $\varphi$ is a reachability property of $M$, then

$$a < b \Rightarrow (M \vDash \varphi \Leftrightarrow Acc_p(M, Cycle) \vDash \varphi).$$

*Yin, Zhuang & Wang (2011)* gives this theorem form forward symbolic reachability analysis, and this is the previous achievement of our working group. We will give its another proof in the view of zone later.

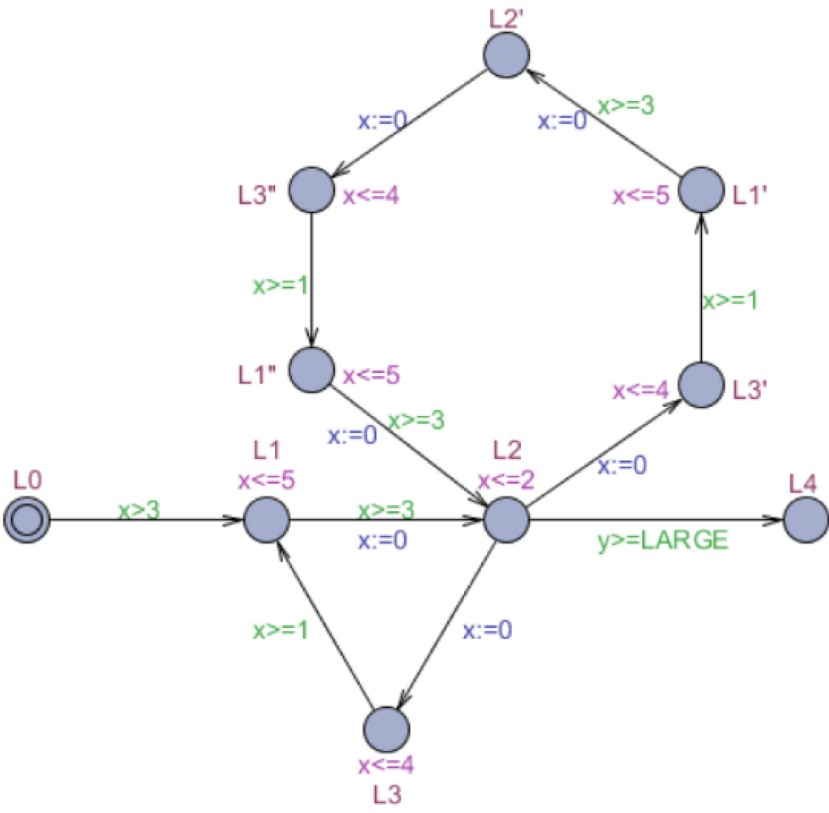

**Figure 2** Accelerated automaton $M_a$ based on appended cycle.

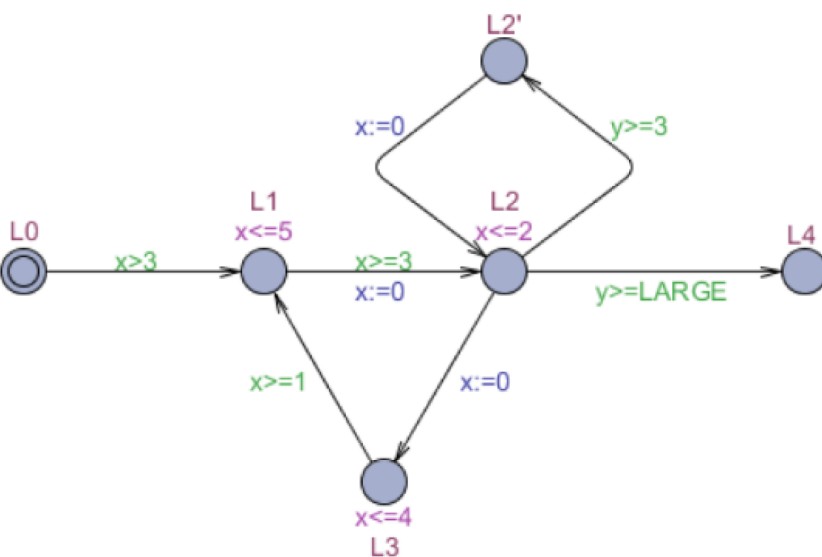

**Figure 3** Accelerated automaton $M_p$ based on parking cycle.

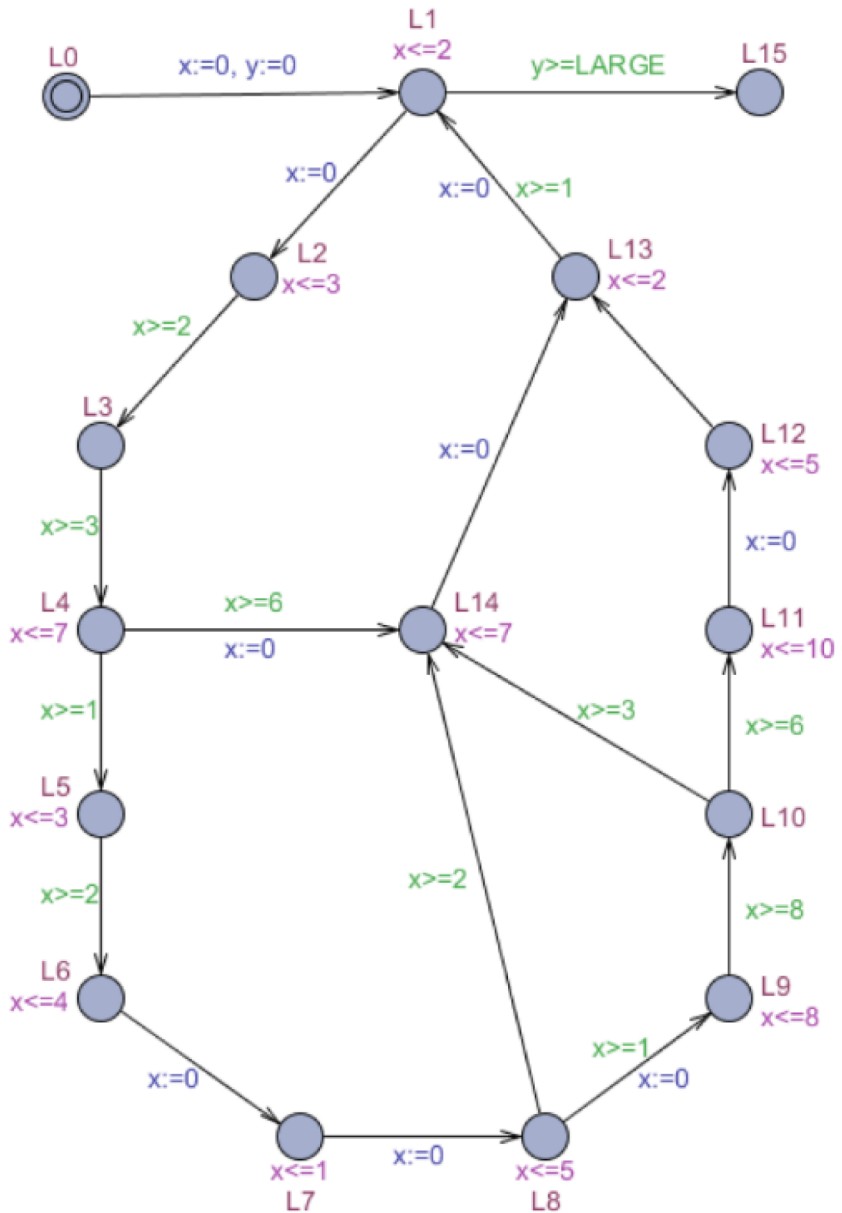

**Figure 4  Timed automaton $M'$.**

## EXACT ACCELERATION OF COMPLEX REAL-TIME SYSTEM MODEL BASED ON OVERLAPPING CYCLE

The appended cycle and parking cycle technologies in exact acceleration apply to a real-time system model with a single acceleratable cycle. For a complex real-time system model (as

shown in Fig. 4), using these two technologies for exact acceleration requires additional states and consumes a large amount of time cost, which influences the acceleration effect.

Figure 4 presents an example of the IoT gateway security system (in *Wang et al., 2018*). The timed automaton $M'$ models a wireless sensor network including a reactive program and external environment. The run-time behavior control several sensors, which can be transformed into clock constrains in UPPAAL. Every location in the cycle represents a sensor model in the IoT system. The cycle's clock is $x$, and clock $y$ controls the execution time. The larger the constant LARGE is, the more slowly the timed automaton $M'$ runs. Using the algorithm described by *Yin, Song & Zhuang (2010)* to identify the acceleratable cycle in $M'$, we obtain four acceleratable cycles whose reset locations are all L1 and share the clock of cycle $x$. For this complex real-time model, we propose a method based on the overlapping cycle for exact acceleration.

**Theorem 3**. Let $M = (C, L, l_0, A, I, E)$ be a timed automaton, and let $Cycle = (E_c, x)$ be an acceleratable cycle of $M$ with a window of $[a, b]$. If $a < b$, there is a positive integer $n$ in the forward symbolic reachability analysis, which leads the reset location to obtain a continuous clock zone after executing the *Cycle* $n$ times.

*Proof.* According to the forward symbolic reachability analysis, the problem of fragments in the acceleratable cycle will inevitably lead to the overlap of clock zones, that is, the appearance of continuous clock zone. If $a < b$, according to the definitions about exact acceleration, the continuous clock zone will be got after several executions of the acceleratable cycle, and the point of the proof is to determine the number of executions.

So, without loss of generality, we might assume that the execution number is a positive integer $n$. Let $n$ be the rounds of *Cycle* execution, and let the interval $[c, d]$ be the clock zone at the reset location before execution of the *Cycle*. At the reset location, the clock zone is continuous from the $(n+1)$th time onward; therefore, the clock zones obtained in the $(n+1)$th time and the $n$th time have an intersection that is

$$(n+1)a+c \leq nb+d \Rightarrow n \geq (a+c-d)/(b-a).$$

Because $b > a$, $d \geq c$, there must be an integer $n \geq a/(b-a)$. So, the number of executions should be at least $\lceil a/(b-a) \rceil$. When the *Cycle* is executed $\lceil a/(b-a) \rceil$ (that is $n$) times, the reset location obtains a continuous clock zone, thereby completing the proof.

**Corollary 1**. If the timed automaton $M$ has an acceleratable cycle with a window of $[a, b]$, $a < b$, then the reset location will obtain a continuous clock zone after executing the *Cycle* at least $a/(b-a)$ times during forward symbolic reachability analysis.

*Proof.* According to Theorem 3, we easily know the reset location obtains a continuous clock zone when the *Cycle* is executed $n$ times. For the integer $n$, we can calculate that $n \geq a/(b-a)$. The proof is completed.

**Corollary 2**. Let the global clock be $y$, and let the maximum value of $y$ before entering the acceleratable cycle be $n_0$. If the timed automaton $M$ has an acceleratable cycle with a window of $[a, b]$, $a < b$, then the reset location will obtain a continuous clock zone when the following condition is satisfied during forward symbolic reachability analysis:

$$y - n_0 \geq a \times a/(b-a).$$

*Proof.* According to Corollary 1, the reset location will obtain a continuous clock zone after executing the acceleratable cycle at least $a/(b-a)$ times. Because the window of the acceleratable cycle is $[a, b]$, the increment of $y$ by executing the acceleratable cycle $a/(b-a)$ times is denoted as $\Delta y \in [a \times a/(b-a), b \times a/(b-a)]$, where $\Delta y = y - n_0$. Therefore, when $y - n_0 \geq a \times a/(b-a)$, the reset location will obtain a continuous clock zone. The proof is completed.

**Corollary 3**. Let the global clock be $y$, and let the maximum value of $y$ before entering the acceleratable cycle be $n_0$. If the timed automaton $M$ has an acceleratable cycle with a window of $[a, b]$, $a < b$, then every location in the acceleratable cycle will obtain a continuous clock zone when $y - n_0 \geq a \times a/(b-a)$.

*Proof.* Because clock $y$ is not the cycle clock, any invariant or guard in the acceleratable cycle will not contain clock $y$ according to Definition 2. Based on the theory of timed automata, clock $y$ exhibits monotonous growth when the acceleratable cycle is executed. Thus, when $y - n_0 \geq a \times a/(b-a)$, per Corollary 2, the reset location begins to obtain a continuous clock zone, indicating that every location in the acceleratable cycle is reachable. At this point, a constant clock zone will also be received by any location in the acceleratable cycle, and the proof is completed.

Next, we will give the new proof of Theorem 2.

*Proof.* Sufficient Condition. The known condition is $a < b$. Because $Acc_p(M, Cycle)$ is obtained by adding a parking cycle to $M$, the timed automaton $M$ can clearly reach any reachable state in original model. The accelerated automaton $Acc_p(M, Cycle)$ can also reach states by executing the same time trace; that is, the state transition system $S(M)$ associated with $M$ is included in the state transition system $S(Acc_p(M, Cycle))$ associated with $Acc_p(M, Cycle)$.

Necessary Condition. The known condition is $a < b$. Let the global clock be $y$. When $y < a \times a/(b-a) + n_0$, the accelerated automaton $Acc_p(M, Cycle)$ does not execute the parking cycle, and reachable states in $Acc_p(M, Cycle)$ are also reachable in the timed automaton $M$. When $y \geq a \times a/(b-a) + n_0$ (according to Corollary 3), every location in the acceleratable cycle of $M$ will obtain a continuous clock zone; that is, after $y \geq a \times a/(b-a) + n_0$ at any time, $M$ can reach any location in the acceleratable cycle and $Acc_p(M, Cycle)$ executes the parking cycle, satisfies the edge guards, and returns to the reset location of any state, which guarantees that $M$ is always reachable.

In summary, when $a < b$, the accelerated automaton $Acc_p(M, Cycle)$ does not change the reachability property $\varphi$ of the timed automaton $M$, and the proof is completed.

According to our theorems and corollaries, we can prove that the exact acceleration method based on the parking cycle is more concise and effective than that based on the appended cycle. On one hand, there is no location invariant in the parking cycle to ensure the clock can stay in this location for acceleration; on the other hand, the parking cycle contains an edge guard to ensure that any location of the acceleratable cycle obtains a continuous clock zone, which provides reachability. In this way, the parking cycle accelerates the search for the symbolic state by controlling acceleration timing and ensures reachability of the timed automaton to realize exact acceleration. The exact acceleration method for the complex real-time model based on an overlapping cycle is an improved

method based on the parking cycle. It attempts to extend the application field of exact acceleration technology to complex real-time model checking to improve efficiency and alleviate state explosion.

**Theorem 4.** Let $M = (C, L, l_0, A, I, E)$ be a timed automaton with several acceleratable cycles. Let $Cycle_i = (E_{c_i}, x)$ be the $i$th acceleratable cycle of $M$ with a window of $[a_i, b_i]$, where $i$ is a non-zero natural number. All acceleratable cycles affect the cycle of clock $x$, and their reset locations are uniform in $l_{reset} \in L$. There is a single acceleratable cycle whose effect is the most effective in obtaining a continuous clock zone than multiple acceleratable cycles.

*Proof.* Let $n_j = \left\lceil \frac{a_j}{b_j - a_j} \right\rceil \times a_j$, $n_k = \left\lceil \frac{a_k}{b_k - a_k} \right\rceil \times a_k$, where $1 \leq j, k \leq i$, and $j, k$ are non-zero natural numbers. Then, $n_j$ and $n_k$ represent the edge guard of the $j$th and $k$th acceleratable cycle, respectively, when adding a parking cycle. If these two acceleratable cycles are executed simultaneously, the edge guard can be expressed as $n_{jk} = \left\lceil \frac{a_j + a_k}{(b_j - a_j) + (b_k - a_k)} \right\rceil \times (a_j + a_k)$. The window of the $i$th acceleratable cycle is $[a_i, b_i]$ as a known condition, where $0 \leq a_j \leq b_j$, $0 \leq a_k \leq b_k$, and there is $a_j + a_k \geq a_j, a_k$.

We make $\frac{a_j}{b_j - a_j} = \frac{U}{V}$, $\frac{a_k}{b_k - a_k} = \frac{X}{Y}$, such that $\frac{a_j + a_k}{(b_j - a_j) + (b_k - a_k)} = \frac{U + X}{V + Y}$. We assume that $\frac{U}{V}$ is smaller, then $\frac{X}{Y} = \frac{U + \mu}{V}$, $\mu \in \mathbb{R}^+$. Therefore, there is $\frac{U + X}{V + Y} = \frac{U + U + \mu}{V + V} = \frac{U}{V} + \frac{\mu}{2V} > \frac{U}{V}$; that is, $\left\lceil \frac{U + X}{V + Y} \right\rceil \geq \left\lceil \frac{U}{V} \right\rceil$ and $\left\lceil \frac{U + X}{V + Y} \right\rceil \geq \min(\left\lceil \frac{U}{V} \right\rceil, \left\lceil \frac{X}{Y} \right\rceil)$. In the positive-number condition, a larger number multiplied by a larger number is either equal to or greater than a smaller number multiplied by a smaller number; therefore,

$$\left\lceil \frac{U + X}{V + Y} \right\rceil \times (a_j + a_k) \geq \min(\left\lceil \frac{U}{V} \right\rceil \times a_j, \left\lceil \frac{X}{Y} \right\rceil \times a_k)$$

which is $n_{jk} \geq \min(n_j, n_k)$.

According to Corollary 1, the reset location will obtain a continuous clock zone after executing the acceleratable cycle, which has a smaller value of $\left\lceil \frac{a_i}{b_i - a_i} \right\rceil \times a_i$, $\left\lceil \frac{a_i}{b_i - a_i} \right\rceil$ times during forward symbolic reachability analysis. This solution is faster than using two acceleratable cycles simultaneously to obtain a continuous clock zone, and it is better than using the larger one.

By extension, when comparing any two acceleratable cycles, a shorter time cycle always obtains a continuous clock zone more quickly. When comparing all acceleratable cycles, we can achieve the most effective acceleratable cycle for exact acceleration. This result indicates that the acceleration effect of a single acceleratable cycle is more effective than that of multiple acceleratable cycles, thereby completing the proof.

**Corollary 4.** Let $M = (C, L, l_0, A, I, E)$ be a timed automaton with several acceleratable cycles. Let $Cycle_i = (E_{c_i}, x)$ be the $i$th acceleratable cycle of $M$ with a window of $[a_i, b_i]$, where $i$ is a non-zero natural number. All acceleratable cycles affect the cycle of clock $x$, and their reset locations are uniform in $l_{reset} \in L$. If $a_i < b_i$, then the acceleratable cycle with the $\min\left(\left\lceil \frac{a_i}{b_i - a_i} \right\rceil \times a_i\right)$ has the best acceleration effect of obtaining a continuous clock zone in the shortest time.

*Proof.* According to Theorem 4, comparing any two acceleratable cycles, a cycle with smaller value of $\left(\left\lceil \frac{a_i}{b_i - a_i} \right\rceil \times a_i\right)$ always obtains a continuous clock zone more quickly.

When comparing all acceleratable cycles, the cycle with the $\min\left(\left\lceil \frac{a_i}{b_i - a_i} \right\rceil \times a_i\right)$ obviously has the most effective acceleration. The proof is completed.

**Definition 6** (Accelerated automaton based on overlapping cycle). Let $M = (C, L, l_0, A, I, E)$ be a timed automaton with $k$ acceleratable cycles, where $L = \{l_0, l_1, \ldots, l_m\}$. $CYCLE = \{Cycle_1, \ldots, Cycle_k | k \in \mathbb{N}^+\}$ denotes the acceleratable cycle set. Let $Cycle_i = (E_{c_i}, x)$ be the $i$th acceleratable cycle with a window of $[a_i, b_i]$, where $0 \leq i \leq k$, $E_c = (e_0, e_1, \ldots, e_{n-1})$, $e_{j_i} = (l_{j_i}, a_{j_i}, \tau_{j_i}, \lambda_{j_i}, l_{(j+1)_i})$. All acceleratable cycles affect the cycle of clock $x$, and their reset locations are uniform in $l_{reset} \in L$. The global clock is $y$, and the maximum value of $y$ before entering the acceleratable cycle is $n_0$. The acceleration of $M$ based on the overlapping cycle is a new automaton $Acc_o(M, CYCLE)$ defined as $(C, L', l_0, A, I', E')$, where

- $L' = L \cup \{l'_{reset}\}$
- $I'(l_h) = I(l_h)$, $0 \leq h \leq m$
- $I'(l'_{reset}) = \varnothing$
- $E' = E \cup \{(l_{reset}, \varnothing, \tau', \varnothing, l'_{reset}), (l'_{reset}, \varnothing, \varnothing, \lambda', l_{reset})\}$, $\tau'$ is $y \geq \min\left(\left\lceil \frac{a_i}{b_i - a_i} \right\rceil \times a_i\right) + n_0$ and $\lambda'$ only contains $x$.

The accelerated automaton $Acc_o(M, CYCLE)$ equals the timed automaton $M$ with the addition of an overlapping cycle at only one reset location, which solves the problem where an exact acceleration cannot be used directly for complex real-time model checking. The overlapping cycle only needs to analyze all windows of every acceleratable cycle in $CYCLE$ for the calculation. We can also avoid using an appended cycle or parking cycle for each acceleratable cycle, greatly reducing the additive symbolic state. Figure 5 depicts the acceleration of $M'$ (Fig. 4) based on an overlapping cycle, named accelerated automaton $M'_o$. Because the timed automaton $M'$ contains four acceleratable cycles, we analyze them separately, discard the deadlocked cycle, and only retain three executable cycles. The deadlocked cycle consists with location $l_1, l_2, l_3, l_4, l_5, l_6, l_7, l_8, l_9, l_{10}, l_{14}, l_{13}, l_1$ in sequence. For further analysis, the windows of acceleratable cycles are calculated as $[7, 18]$, $[6, 16]$, and $[13, 24]$. By taking the minimum value of $\left\lceil \frac{a_i}{b_i - a_i} \right\rceil \times a_i$, we can obtain the overlapping cycle's entry condition, which is $y \geq 6$.

**Theorem 5.** Let $M = (C, L, l_0, A, I, E)$ be a timed automaton with several acceleratable cycles. Let $Cycle_i = (E_{c_i}, x)$ be the $i$th acceleratable cycle of $M$ with a window of $[a_i, b_i]$, where $i$ is a non-zero natural number. If $x$ is reset on edge $e_0$, then the subsequent states of $src(e_0)$ reachable by multiple acceleratable cycles in $M$, are reachable by exactly one execution of the overlapping cycle in $Acc_o(M, CYCLE)$.

*Proof.* For a certain acceleratable cycle, its window is set as $[a', b']$. According to Theorem 1, when $3a' \leq 2b'$, the appended cycle does not change the subsequent reachability of the reset location $src(e_0)$ in $M$. According to Theorem 2, when $a' \leq b'$, the parking cycle does not change the subsequent reachability of the reset location $src(e_0)$ in $M$. In the case of multiple acceleratable cycles superimposing on the same location in a complex real-time system, the subsequent reachability of the reset location $src(e_0)$ in $M$ can be guaranteed to remain unchanged if any part of the acceleratable cycle is processed with exact acceleration.

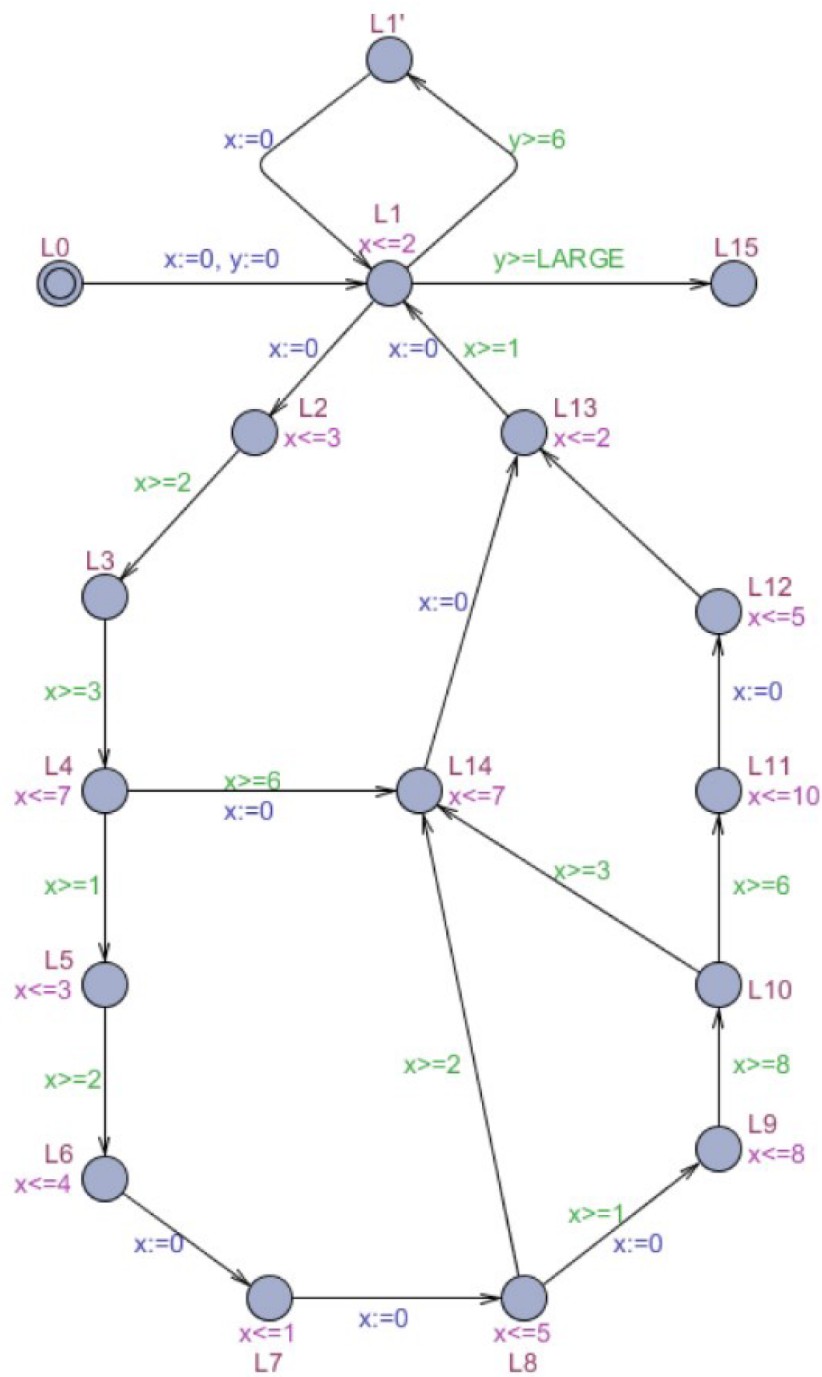

**Figure 5** Automaton $M'_o$: acceleration of $M'$ based on an overlapping cycle.

**Table 2  Runtime data comparing $M'$ and its accelerated versions $M_{a'}$, $M_{p'}$ and $M_{o'}$.**

| | | LARGE | | | | | |
|---|---|---|---|---|---|---|---|
| | | $10^4$ | $10^5$ | $10^6$ | $10^7$ | $10^8$ | $10^9$ |
| $M'$ | Mem (KB) | 27,020 | 26,892 | 27,392 | 27,544 | 27,952 | 28,572 |
| | Time (s) | 0.032 | 0.256 | 1.688 | 12.045 | 120.333 | 1,191.025 |
| $M_a'$ | Mem (KB) | 28,328 | 29,220 | 30,468 | 31,688 | 32,508 | 34104 |
| | Time (s) | 0.007 | 0.008 | 0.008 | 0.008 | 0.009 | 0.009 |
| $M_p'$ | Mem (KB) | 27,184 | 27,384 | 27,788 | 28,060 | 28,208 | 29,276 |
| | Time (s) | 0.005 | 0.004 | 0.003 | 0.003 | 0.003 | 0.004 |
| $M_o'$ | Mem (KB) | 27,164 | 27,036 | 27,472 | 27,824 | 27,988 | 28,788 |
| | Time (s) | 0.002 | 0.002 | 0.002 | 0.003 | 0.002 | 0.002 |

According to Theorem 4, the accelerable cycle is more effective for obtaining a continuous clock zone at reset location $src(e_0)$ than multiple acceleratable cycles. In particular, according to Corollary 4, if $a_i < b_i$, then the exact acceleration based on overlapping cycle can obtain the continuous clock zone in the shortest time. The proof is completed.

This theorem ensures the effectiveness of acceleration. For a single acceleratable cycle, if all states are reachable by more than one execution of the acceleratable cycle, then exactly only one execution of the acceleratable cycle of the appended cycle or parking cycle can guarantee reachability of all states in the accelerative automaton. The complex real-time model checking differs from the exact acceleration of a single acceleratable cycle. In depth-first forward symbolic reachability analysis, it is necessary to verify whether subsequent states are reachable in priority while ignoring the breadth-first search within cycles.

In our case study of an IoT gateway security system (*Wang et al., 2018*), the control center must complete a security process and distribute it to each sensor node. Once a self-organizing sensor network completes the process, it can respond to the command of the control center in a timely manner. The control center can perform subsequent operations after receiving feedback regardless of whether other sensor nodes can complete the process. Hence, the security system must ensure its subsequent reachability regardless of who completes the process. This approach accelerates the search of subsequent states, thus avoiding time and space consumption caused by superposition of acceleratable cycles.

## EXPERIMENTAL RESULTS

To verify the validity of the exact acceleration method based on an overlapping cycle for complex real-time model checking, we collected runtime data, including memory consumption and verification time, from the timed automaton $M'$. We also gathered runtime data from the accelerated automata $M_a'$, $M_p'$, and $M_o'$, which use the appended cycle, parking cycle, and overlapping cycle, respectively. We employed the model-checking tool UPPAAL with a depth-first search order to verify whether location L15 was reachable, which can give the time and memory consumption in verification automatically. Experimental results are displayed in Table 2.

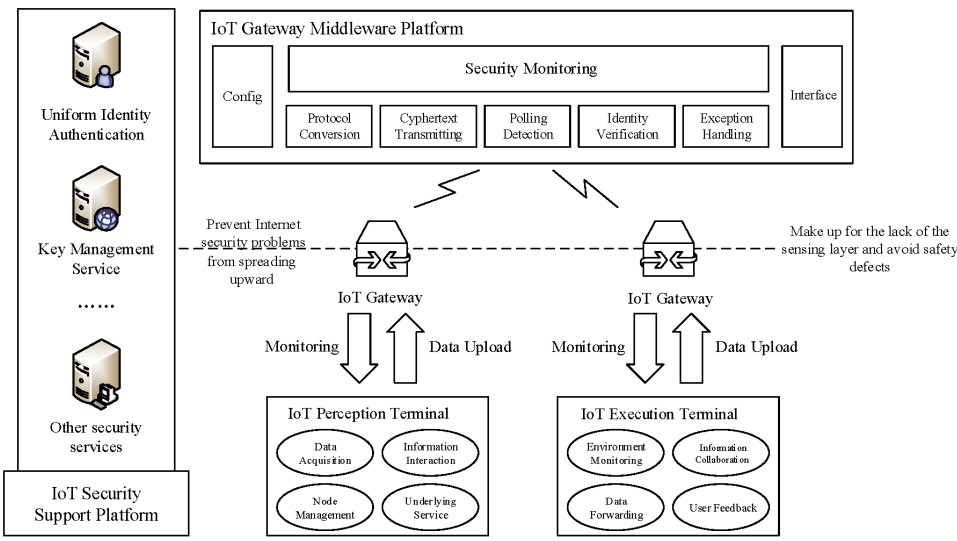

**Figure 6 Technical framework of IoT gateway security system.**

Results show that the time consumption of $M'$ increased with exponential growth of LARGE at a rate of nearly ten times without using exact acceleration. The memory consumption of $M'$ increased slightly because no additional locations were added. The accelerated automaton $M_a'$ used an appended cycle, which reduced the time consumption, reflecting the advantages of exact acceleration. However, due to a large number of additional locations, the memory consumption of $M_a'$ increased dramatically. The accelerated automaton $M_p'$ used a parking cycle to reduce the time consumption further compared to $M_a'$. The fixed length of the parking cycle reduced the number of additional locations compared to the appended cycle; accordingly, the memory consumption was much lower for $M_p'$ than for $M_a'$ but slightly higher than for $M'$. The accelerated automaton $M_o'$ that used the proposed overlapping cycle exhibited minimal time consumption and only required the addition of a single, fixed-length location for the complex real-time model. The memory consumption of $M_o'$ was close to that of $M'$, far less than that of $M_a'$, and slightly better than that of $M_p'$. We can explain the time and memory consumption of $M_o'$ by Theorem 5. The depth-first search order ensures that the overlapping cycle accelerates exploration before complete exploration of all accelerable cycles.

A less theoretical study case involves model verification of the IoT gateway security system (*Wang et al., 2018*). The exact acceleration method based on the overlapping cycle was successfully applied in this case, significantly improving verification efficiency. The technical framework of the IoT gateway security system is illustrated in Fig. 6.

An essential technology in the IoT gateway security system is the time-stamped advanced encryption standard algorithm. By introducing a timestamp into the key expansion phase, the round key can be dynamically updated with change over time to realize a cipher text change that ensures the security of confidential information. Due to the introduction of a timestamp, the system generates acceleratable cycles when modeled as timed automata.

Multiple acceleratable cycles are overlaid on the same location in particular scenarios, which requires overlapping cycle technology for exact acceleration.

Our theory can be used to simulated the parallel execution of processes and idle cycles. However, the presence of urgent locations and synchronous channels may disturb exact acceleration. For example, if broadcast channel coordination occurs in an urgent location, the multi-party response of the broadcast should be completed before the next state location can be migrated. The execution time of the response process is not controlled by the cycle control program; thus, it is not appropriate to simply use exact acceleration technology for acceleration; the space–time loss of using acceleration technology should be compared to the broadcasting response. However, exact acceleration technology can often handle urgent locations and synchronous channels. The following case of an IoT gateway security system also involves these situations. As no extra time interference exists within the whole acceleration process, the exact acceleration technology can finally be successfully applied to system modeling and verification.

The accelerated automaton based on an overlapping cycle is an approximation that can be adapted to verify the accuracy of invariance and reachability properties. Consider the processes in Fig. 7; these processes model a top architecture and a middleware control program consisting of locations, edges, and channels.

The IoT gateway runs from the Start location, reads configuration information and performs gateway identity authentication. The underlying unified authentication service is invoked through channel StartAC for security authentication, and *GatewayStatus* is returned after authentication. If *GatewayStatus* =true, then the system enters the location EnterMiddle and transforms to the polling module of the middle layer through the StartMiddle channel. For the middle-layer polling module, polling begins through the StartMiddle channel, and the top-layer main module is returned by the FinishMiddle channel. Location CheckCategory controls whether polling logic ends up at a perception terminal or an execution terminal, which each have different processing methods. In the two polling processes, the underlying security service is invoked through the synchronization channel according to different requirements. The specific process can be interpreted by the meaning of state locations, synchronization channels, and variables as described by *Wang et al. (2018)*. In particular, clock constraints are added during the stage of waiting for timing and the stage of keeping the equipment running. The top-layer main module model and middle-layer polling module model constitute the general framework of the IoT gateway security system. Security implementation depends on the underlying security service modules ultimately, hence it is necessary to improve model construction of the underlying security service modules. For complex system modeling with underlying services, we apply the exact acceleration method proposed in this paper, which can effectively improve the verification speed.

To demonstrate the effect of exact acceleration, we checked all security properties of the IoT gateway security system model in Fig. 7. Several examples are presented below.

(1) A[] not deadlock

Property 1 is used to check deadlock and ensure all state locations will be reachable.

(2) E<>Top.CheckGS

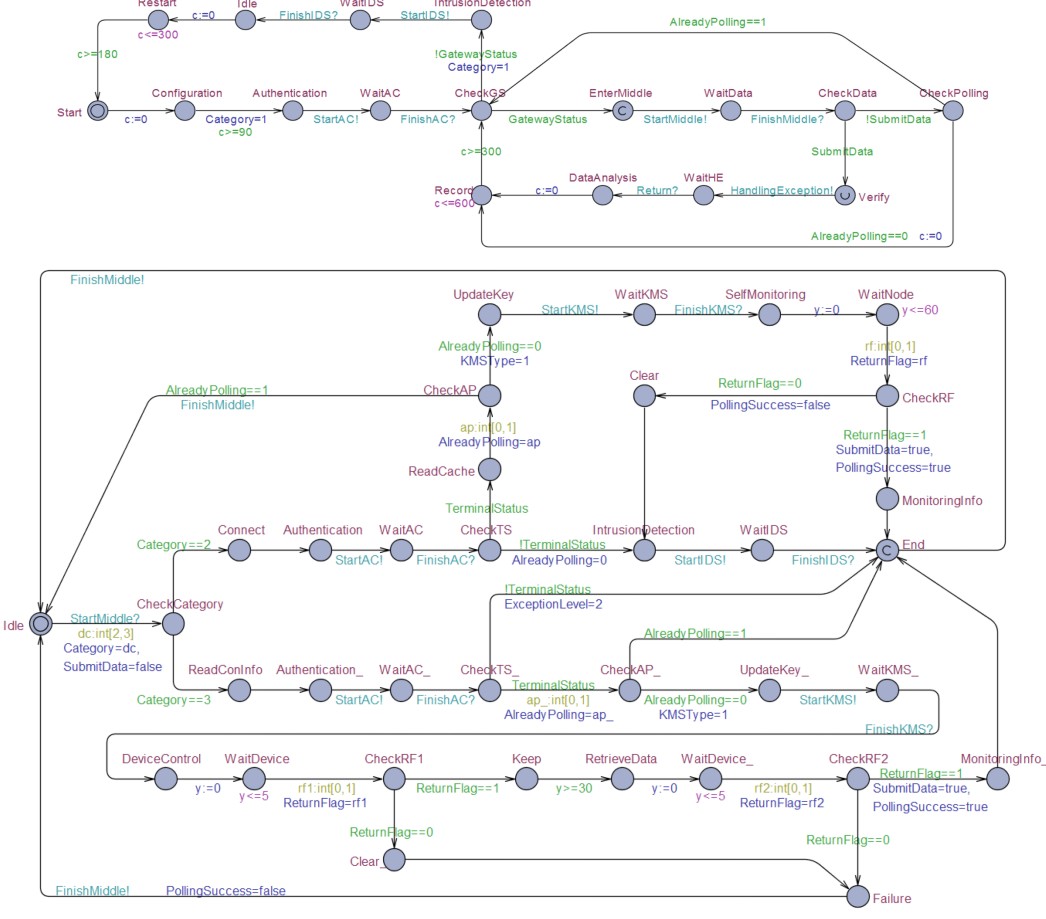

**Figure 7** **Part of the IoT gateway security system model.**

(3) E<>Top.EnterMiddle imply Middle.CheckCategory

Properties 2 and 3 are used to explore part of the state space. The truth of these two properties indicates that the implementation accelerated model is an exact acceleration with the overlapping cycle.

(4) A[] Top.Restart imply $c<=300$

(5) A[] Top.Record imply $c<=600$

(6) A[] Middle.RetrieveData imply Middle.$y>=30$

(7) A[] Middle.WaitDevice imply Middle.$y<=5$

Properties 4–7 are examined in terms of whether subsequent states of the reset location are reachable. Clock $c$ is a global clock and clock $y$ is used to model the duration of one process.

We measured time and memory consumption and explored states for these properties. The IoT gateway security system was modeled as a timed automaton $M_{IoT}$, and the acceleration of $M_{IoT}$ with overlapping cycles was modeled as an automaton $M_{IoT_o}$. We used model checkers UPPAAL and KRONOS to verify security system properties automatically, such as confidentiality, availability, and authenticity in parallel processes. KRONOS is able

**Table 3** Runtime data comparing $M_{IoT}$ and $M_{IoT_o}$.

|  | Explored States | Time(s) | Memory(KB) |
|---|---|---|---|
| $M_{IoT}$ | 108,302 | 71.151 | 29,660 |
| $M_{IoT_o}$ | 47,545 | 1.049 | 30,840 |

**Table 4** Comparing the performance of different exact acceleration techniques for large-scale IoT systems.

| System states-scale | Exact acceleration technique | Verification time(s) |
|---|---|---|
| $10^4$ | Appended cycle | 277.860 |
| $10^4$ | Parking cycle | 0.893 |
| $10^4$ | Overlapping cycle | 0.015 |
| $10^5$ | Appended cycle | $\infty$ |
| $10^5$ | Parking cycle | 72.218 |
| $10^5$ | Overlapping cycle | 1.020 |
| $10^6$ | Parking cycle | 364.720 |
| $10^6$ | Overlapping cycle | 43.292 |
| $10^7$ | Parking cycle | $\infty$ |
| $10^7$ | Overlapping cycle | 409.132 |

to complete the statistics of the state scale traversed by the whole verification process. It makes up for the fact that UPPAAL can't do this. Table 3 lists the experimental results.

On the premise of guaranteeing the security of IoT gateway system, a large number of underlying services and various applications can be embedded in the system framework. At this time, the security requirements of IoT gateway system are mainly for various new access services, and the framework security of the gateway itself can be maintained by its own mechanism. After access to a large number of services and applications, the original model will become complex, concurrent, real-time with large-scale. The verification of the system needs to be processed by the exact acceleration method based on overlapping cycle.

With the increase of the number of access services, the system model becomes more and more complex, and the scale of access number greatly affect the efficiency of model verification. Appended cycle and parking cycle methods are more suitable for single accelerating cycle scenarios. In this complex scenario, when the number of services reaches a certain level, the acceleration process may not be completed. According to the change of the number of access services, Table 4 gives the comparison of the acceleration effects of different exact acceleration methods (from the perspective of time).

The results show that for complex real-time systems, the acceleration efficiency of overlapping cycle is much higher than that of appended cycle and parking cycle, and the verification can still be completed when the state scale reaches $10^7$ with proposed method. So, the exact accelerating technology substantially reduced the time required for verification in complex real-time model checking. Overlapping cycle acceleration demonstrated the highest efficiency compared to the appended cycle and parking cycle. In the simple example of automaton $M'$ in Fig. 4, 55 additional locations were required when using the appended

cycle, much higher than the number of locations in the original model. Although the appended cycle reduced verification time, it increased the difficulty of adding locations to the model in an early stage. When many acceleratable cycles were stacked in the same reset position, more than one location needed to be added to $M'$ when the parking cycle was used, although the length of the parking cycle was fixed. The parking cycle was neither simpler nor faster than the overlapping cycle, and its previous calculation was larger than that of the overlapping cycle.

With the exception of this IoT case, our approach can be applied to other scenarios, such as security validation of blockchain smart contracts. The complete code and UPPAAL model can be found at https://github.com/iegqwang/UPPAAL.

## CONCLUSIONS

To solve the fragmentation problem for complex real-time model checking, we propose an exact acceleration method based on an overlapping cycle, which is an application scenario extension of parking-cycle technique, to accelerate forward symbolic reachability analysis. Compared with the appended cycle or parking cycle for exact acceleration, the proposed method can be applied to the model acceleration of large-scale complex real-time systems and only requires the addition of a single, fixed-length location to the system's timed automaton model. The addition of an overlapping cycle introduces far fewer symbolic states than using either an appended cycle or parking cycle. Rather than relying on windows of acceleratable cycles, the proposed accelerated automaton model is more straightforward and reduces the space–time overhead of exact acceleration.

Two aspects warrant exploration in future research. First, we must continue to study the algorithm for the acceleratable cycle, try to simplify the original automaton model, guarantee its original property, and rapidly identify the deadlock. Second, we plan to develop a simple exact acceleration automatic checking platform that can consider other practical conditions such as action transitions, urgent locations, and synchronous channels to solve actual modeling problems more efficiently.

### Funding
This work was supported by the Key Program of the National Natural Science Foundation of China (No. U1604262), the Key Scientific Research Project of Higher Education of Henan (No. 19A520003, 18A520006 and 17A520057), and the Key R&D and Promotion Project in Science and Technology of Henan (No. 182102210189). The funders had no role in study design, data collection and analysis, decision to publish, or preparation of the manuscript.

### Grant Disclosures
The following grant information was disclosed by the authors:
Key Program of the National Natural Science Foundation of China:  U1604262.

Key Scientific Research Project of Higher Education of Henan: 19A520003, 18A520006, 17A520057.
Key R&D and Promotion Project in Science and Technology of Henan: 182102210189.

## Competing Interests

The authors declare there are no competing interests.

## Author Contributions

- Guoqing Wang conceived and designed the experiments, performed the experiments, performed the computation work, prepared figures and/or tables, and approved the final draft.
- Lei Zhuang conceived and designed the experiments, performed the computation work, prepared figures and/or tables, and approved the final draft.
- Yu Song analyzed the data, authored or reviewed drafts of the paper, and approved the final draft.
- Mengyang He performed the experiments, analyzed the data, prepared figures and/or tables, and approved the final draft.
- Ding Ma performed the experiments, performed the computation work, authored or reviewed drafts of the paper, and approved the final draft.
- Ling Ma analyzed the data, authored or reviewed drafts of the paper, and approved the final draft.

## Data Availability

Codes are available at GitHub: https://github.com/iegqwang/UPPAAL.

## Supplemental Information

Supplemental information for this article can be found online at http://dx.doi.org/10.7717/peerj-cs.272#supplemental-information.

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
