# Peer review of "Exact acceleration of complex real-time model checking based on overlapping cycle"

_PeerJ Computer Science, doi:10.7717/peerj-cs.272_

## Round 0.1 · original submission · Major Revisions

Dear authors,

your paper has been reviewed by two expert reviewers. They find the presented results interesting, but currently not completely understandable nor attestable. Therefore, you are asked to carefully revise your paper to take all comments from the reviewers into account. In particular, you need to address the following pertinent comments:

- make it crystal clear what is the novelty of the paper with respect to the literature (Reviewer 1)

- carefully revise your definitions and results, in particular Theorem 2 on which other results depend (Reviewers 1 and 2)

- expand on the experimental evaluation to eplain which methodology and tools were used (Reviewer 2)

Without this, it is not possible to properly assess your contribution (the 'multiple overlapping cycles' method) neither in terms of correctness nor in terms of efficiency with respect to existing methods from the literature.

·

Basic reporting

This paper develops an extension of the parking-cycle acceleration
technique for timed automata to consider multiple overlapping cycles.
The extension is shown to be exact with respect to reachability
properties, and supporting experiments show that it performs superior
to the standard parking-cycle technique, and also to the
appended-cycle method, and permits application of timed-automata model
checking to systems which hitherto were too complex for standard
timed-automata model checkers.

The paper is reasonably well-written and carefully covers previous
work on accelerating cycles as well as possible applications to
real-world IoT systems. However, the papers is imprecise in some
important places, hence I recommend a major revision. I do believe
that all results in the paper are correct, but in order for me to be
sure, some technicalities need to be stated much more precisely, see
below.


Major trouble:

1.
Both your abstract and your introduction and conclusion give the
impression that you here introduce a completely new technique for
accelerating cycles. I would dispute this: in my opinion, you extend
the parking-cycle technique to take into account multiple overlapping
cycles which share their reset location. Now this is a good
contribution to TA model checking; I do not dispute this. But it
would help to pur your work in context to know from the outset that
you extend the parking-cycle technique rather than introducing a
completely new technique.

Please change your abstract, introduction and conclusion to indicate
that you extend the parking-cycle technique.

2.
Def.3 is not at all clear. Please define precisely what are the
valuations v_i, v_i' in your trace. Define precisely the notion of
"total delay". What does it mean to "adjust the delays" in T_r(E_c)?

This defines the window of an acceleratable cycle, which is basic for
all which is to come; rather important to get this straight...

3.
In Thm.2 you introduce a notion of "obtain[ing] a continuous clock
zone" which you do not define precisely. I don't understand what it
means, and everything after seems to depend on this notion, so this
inhibits me from checking all proofs after Thm.2.

Please define carefully what it means to "obtain a continuous clock
zone".


Minor details (numbers are line numbers):

49f: I did not understand "however, these ... test completeness."

62: "...when a time*d*"

114ff: You say twice here that "Larsen proposed/modeled", citing
papers with multiple authors. Now Kim is a nice guy and does a lot of
important work; but you should give his co-authors credit and rather
say "Boudjani et al proposed" and "Chadli et al modeled".

174: "... cycle *may be* exited."

187: "... can be present*ed* as ..."

188: "... form*ula* ..."

197: What do you mean "clock y grows more gradually"?

200: "... when the time*d* ..."; what do you mean "detects a symbolic
model"?

212f: You use the same "c" in {x\le c} and {x\ge c}; but they should
be different

253: "... pro*ved* ..."

259: You have not introduced the notion "global clock". Please do so

270: "... permi*tted* to"

275: The reader would expect here that an analogue of Thm.1 holds for
the parking-cycle enhanced timed automaton. This is Thm.3 below, but
maybe you should announce it here already

331ff: I'm a bit surprised that you don't refer to Yin, Zhuang, Wang
2011 for this proof. You say that parking cycles have been introduced
in that paper; but why, then, is Thm.3 not already in that paper? What
worth is the parking-cycle technique without a proof that it is exact?

336f: What do you mean "satisfy any reachable state"?

356: "... timing /of/ and ..." (delete "of")

361ff: Some imprecision here: what does it mean to be "more
effective"? Also, you should rather say "there is a cycle which is
most effective" or make precise which

372: What do you mean "a_j+a_i\ge \forall(a_j, a_k)"?

419f: Please say which cycle is deadlocked and discarded

495: "automat*a* ..."

486: "... when model*ed* as ..."

504-535: It is not necessary to spend so much space on explaining the
IoT example; this is not the main focus of the paper. Please shorten

537: "... be*l*ow." (only one "l")

586: It is not clear from reading your paper whether your analysis of
overlapping cycles has been automated or whether you do this
manually. Please specify

Experimental design

no comment

Validity of the findings

no comment

Additional comments

no comment

Reviewer 2 ·

Basic reporting

## Summary
The paper is concerned with a problem in model checking of timed
automata containing cycles in the discrete graph. The problem is that
the cycles results in a fragmented zone-graph - this paper tries to
syntactically the timed
automata such that reachability properties are preserved while
minimising fragmentation. In the litterature this kind of
fragmentation reduction is called acceleration and has perviously been
studied by numerous authors.

- Larsen and Hendriksens method called appended cycles basically
identifies cycles that can be accelerated, unrolls the cycles a
number of times abd appends the "unrolled" version of the cycles to
the automaton.

- Yin, Zhuang and Wangs methods is called parking cycle. In this methd
the cycle is not unrolled - instead it is identified when the zones
visited by the cycles becomes continuous. At this point a cycle is
appended to the cycle in which time can delay infinitely long (this
is the parking cycle).

The above methods focuses on accelerating just one cycles and
overlooks the fact that sometimes multiple cycles can be accelerated
from the same location. This special case is considered in this
paper. In particular their results indicate that accelerating just one
cylce (with the parking cycle technique) is better than accelerating
all of them.

The paper is concluded with an experimental section documenting the
effects of acceleration involving both Uppaal and Kronos.

## Structure of the paper
The paper is fairly well-structured first discussing the state of the
art, and then the results of the paper itself. One big annoyance of
mine is that all of the figure and tables are at the end of the paper
(even after the appendix). I am not sure if this is caused by the
template of PeerJ or the authors selected this - anyway I do find
it annoying. Furthermore, given the fact that the figures are all at
the end of the paper, the authors ought to provide proper references
(i.e. Figure 1 instead of Timed Automaton M).

It's not always completely clear when a proof is finished. (no visual
aid to separate the proof from the running text)

## Language
The language of the paper is okay, but definitely not perfect. As
an example I needed subtstantial time for understanding what the
"window of a cycle" ( Definition 3) was, simply because I did not
understand the explanation. To my understanding, it is the minimal
and maximal time it may take to pass through a cycle.

Experimental design

The experimental evaluation consists of the evaluation of a small
toy-example for which results show that the acceleration techniques
does indeed reduce the time verification time. I can however not see
this very clear indication of higher memory consumption of the
appended cycles technique. In fact, all of the techniques (appended
cycle, parking cycle and this papers technique) use around 30000K.

A larger IOT model was investigated as well. For this model it is
again revealed that acceleration is quite nice for reducing the time
and in particular that the authors technique is better than the
parking and appending cycle techniques. Again the authors talks about
the memory consumption but results show that with and without
acceleration the verification uses around 30000K. THis is strange and
I would almost think, that something has gone wrong with the memory
measurement.

A curiousity in regards to the evaluation is, that the paper says the
IOT case study was conducted using UPPAAL and Kronos. I find this
curious for two reasons:
1. To my knowledge Kronos has been discontinued for many years, and
2. The models presented in the paper are all Uppaal models - likewise
are the models in the supplementary material all uppaal models.

Validity of the findings

## Correctness
In regards to whether the results are correct, I cannot tell. The
proofs seem okay most of them, however they also depend on
concepts that are not really defined in the paper (e.g. continuous
clock-zone). Furthermore, quite a number of the proofs depend on
Theorem 2 which might be true, but the proof of Theorem 2 proves its
correctnes by assuming it is correct (see detailed comments later).



## Theorem 2
As noted earlier this proof appears to be proven by assuming the
theorem is true. According to the Theorem it wants to prove that if
you have an acceratable cycle with window [a,b] with a < b then
there exists a number n such that after n executions of the cycles,
the clock zone becomes continuous. The proof starts out by saying,
"Let n be the rounds of cycle executions when the reset location
obtains a continuous clock zone....", but we want to prove that n
exists (thus the authors assume the theorem to be true while proving
it).

Additional comments

## Overall Evaluation
Presently I do not see this paper as publishable. It does detail
interesting results, but I am unable to validate the correctnes of
proofs - mainly because they depend on Theorem 2 which I do not trust
the proof of. Furthermore, the experimental evaluation is lacking in
regards to explaining its methodology (which tools (Kronos/Uppaal) was actually used,
how was memory measured etc.)

---

## Round 0.2 · Minor Revisions

Dear authors,

Your revised paper has again been reviewed by two expert reviewers. They acknowledge that the paper has improved a lot. However, one of the reviewers is still not convinced of the proof of Theorem 3. He/she does believe the statement. Therefore, please undertake all minor revisions as suggested by the reviewers and, in particular, carefully revise the proof of Theorem 3 based on the doubts raised by reviewer 2.

·

Basic reporting

This paper develops an extension of the parking-cycle acceleration
technique for timed automata to consider multiple overlapping cycles.
The extension is shown to be exact with respect to reachability
properties, and supporting experiments show that it performs superior
to the standard parking-cycle technique, and also to the
appended-cycle method, and permits application of timed-automata model
checking to systems which hitherto were too complex for standard
timed-automata model checkers.

The paper is reasonably well-written and carefully covers previous
work on accelerating cycles as well as possible applications to
real-world IoT systems. Except for some minor remarks below, the
paper can be accepted.

l.26: "... of *the* parking-..."

l.30f: I did not understand the sentence starting "For the". (How
"can [it] accelerate the [...] concurrent states"?) Please
reformulate.

l.49f: Same here: please reformulate the part of the sentence after
"these methods".

l.62: "... which *can* cause ..."

l.101: "... optimizes *the verification of* the LEGO ..."

l.121: Please remove the line break before "theory".

l.138: "... introduces /the/ timed" (remove "the")

l.139: "..., and *the* theory ..."

l.150: T(C) does not "contain[...] all clock variables", please
reformulate.

l.161: "... is the *set of* initial location*s*. ..."

l.162: "... clock constraint/s/"

l.199: What do you mean "the timed automaton detects a symbolic model
wth cycles"? Please clarify.

l.248: You put I'(l_0')=\emptyset, but that's supposed to be a clock
constraint. Do you mean "true"?

l.268f: Same here, both for I'(l_0') and for the guard on the second
added edge.

l.276: "... accelerated automat*a* ..."

l.293f: I didn't understand the sentence starting "The model". Please
reformulate.

l.376: Please remove the first "l_reset".

l.376f: Please clarify your statement by saying "There is a single
acceleratable cycle whose effect is ...".

l.383: The condition "a_j+a_k\ge \forall(a_j, a_k)" is mathematically
meaningless. Please reformulate.

l.404: Please remove the first "l_reset".

l.415: Please remove the first "l_reset".

l.421f: Again some strange occurrences of \emptyset. You probably
mean I'(l_reset')=true, but on the edges in l.422 you twice put a
*label* \emptyset. Do you mean \epsilon here? (There is one more
\emptyset which should be "true", too.)

l.472: "..., from *the* timed ... from *the* accelerated"

l.503: What do you mean "overcome the parallel composition"? Please
clarify.

l.566: What do you mean "the number and scale of access number"?
Please clarify.

Experimental design

This paper develops an extension of the parking-cycle acceleration
technique for timed automata to consider multiple overlapping cycles.
The extension is shown to be exact with respect to reachability
properties, and supporting experiments show that it performs superior
to the standard parking-cycle technique, and also to the
appended-cycle method, and permits application of timed-automata model
checking to systems which hitherto were too complex for standard
timed-automata model checkers.

The paper is reasonably well-written and carefully covers previous
work on accelerating cycles as well as possible applications to
real-world IoT systems. Except for some minor remarks below, the
paper can be accepted.

l.26: "... of *the* parking-..."

l.30f: I did not understand the sentence starting "For the". (How
"can [it] accelerate the [...] concurrent states"?) Please
reformulate.

l.49f: Same here: please reformulate the part of the sentence after
"these methods".

l.62: "... which *can* cause ..."

l.101: "... optimizes *the verification of* the LEGO ..."

l.121: Please remove the line break before "theory".

l.138: "... introduces /the/ timed" (remove "the")

l.139: "..., and *the* theory ..."

l.150: T(C) does not "contain[...] all clock variables", please
reformulate.

l.161: "... is the *set of* initial location*s*. ..."

l.162: "... clock constraint/s/"

l.199: What do you mean "the timed automaton detects a symbolic model
wth cycles"? Please clarify.

l.248: You put I'(l_0')=\emptyset, but that's supposed to be a clock
constraint. Do you mean "true"?

l.268f: Same here, both for I'(l_0') and for the guard on the second
added edge.

l.276: "... accelerated automat*a* ..."

l.293f: I didn't understand the sentence starting "The model". Please
reformulate.

l.376: Please remove the first "l_reset".

l.376f: Please clarify your statement by saying "There is a single
acceleratable cycle whose effect is ...".

l.383: The condition "a_j+a_k\ge \forall(a_j, a_k)" is mathematically
meaningless. Please reformulate.

l.404: Please remove the first "l_reset".

l.415: Please remove the first "l_reset".

l.421f: Again some strange occurrences of \emptyset. You probably
mean I'(l_reset')=true, but on the edges in l.422 you twice put a
*label* \emptyset. Do you mean \epsilon here? (There is one more
\emptyset which should be "true", too.)

l.472: "..., from *the* timed ... from *the* accelerated"

l.503: What do you mean "overcome the parallel composition"? Please
clarify.

l.566: What do you mean "the number and scale of access number"?
Please clarify.

Validity of the findings

This paper develops an extension of the parking-cycle acceleration
technique for timed automata to consider multiple overlapping cycles.
The extension is shown to be exact with respect to reachability
properties, and supporting experiments show that it performs superior
to the standard parking-cycle technique, and also to the
appended-cycle method, and permits application of timed-automata model
checking to systems which hitherto were too complex for standard
timed-automata model checkers.

The paper is reasonably well-written and carefully covers previous
work on accelerating cycles as well as possible applications to
real-world IoT systems. Except for some minor remarks below, the
paper can be accepted.

l.26: "... of *the* parking-..."

l.30f: I did not understand the sentence starting "For the". (How
"can [it] accelerate the [...] concurrent states"?) Please
reformulate.

l.49f: Same here: please reformulate the part of the sentence after
"these methods".

l.62: "... which *can* cause ..."

l.101: "... optimizes *the verification of* the LEGO ..."

l.121: Please remove the line break before "theory".

l.138: "... introduces /the/ timed" (remove "the")

l.139: "..., and *the* theory ..."

l.150: T(C) does not "contain[...] all clock variables", please
reformulate.

l.161: "... is the *set of* initial location*s*. ..."

l.162: "... clock constraint/s/"

l.199: What do you mean "the timed automaton detects a symbolic model
wth cycles"? Please clarify.

l.248: You put I'(l_0')=\emptyset, but that's supposed to be a clock
constraint. Do you mean "true"?

l.268f: Same here, both for I'(l_0') and for the guard on the second
added edge.

l.276: "... accelerated automat*a* ..."

l.293f: I didn't understand the sentence starting "The model". Please
reformulate.

l.376: Please remove the first "l_reset".

l.376f: Please clarify your statement by saying "There is a single
acceleratable cycle whose effect is ...".

l.383: The condition "a_j+a_k\ge \forall(a_j, a_k)" is mathematically
meaningless. Please reformulate.

l.404: Please remove the first "l_reset".

l.415: Please remove the first "l_reset".

l.421f: Again some strange occurrences of \emptyset. You probably
mean I'(l_reset')=true, but on the edges in l.422 you twice put a
*label* \emptyset. Do you mean \epsilon here? (There is one more
\emptyset which should be "true", too.)

l.472: "..., from *the* timed ... from *the* accelerated"

l.503: What do you mean "overcome the parallel composition"? Please
clarify.

l.566: What do you mean "the number and scale of access number"?
Please clarify.

Additional comments

This paper develops an extension of the parking-cycle acceleration
technique for timed automata to consider multiple overlapping cycles.
The extension is shown to be exact with respect to reachability
properties, and supporting experiments show that it performs superior
to the standard parking-cycle technique, and also to the
appended-cycle method, and permits application of timed-automata model
checking to systems which hitherto were too complex for standard
timed-automata model checkers.

The paper is reasonably well-written and carefully covers previous
work on accelerating cycles as well as possible applications to
real-world IoT systems. Except for some minor remarks below, the
paper can be accepted.

l.26: "... of *the* parking-..."

l.30f: I did not understand the sentence starting "For the". (How
"can [it] accelerate the [...] concurrent states"?) Please
reformulate.

l.49f: Same here: please reformulate the part of the sentence after
"these methods".

l.62: "... which *can* cause ..."

l.101: "... optimizes *the verification of* the LEGO ..."

l.121: Please remove the line break before "theory".

l.138: "... introduces /the/ timed" (remove "the")

l.139: "..., and *the* theory ..."

l.150: T(C) does not "contain[...] all clock variables", please
reformulate.

l.161: "... is the *set of* initial location*s*. ..."

l.162: "... clock constraint/s/"

l.199: What do you mean "the timed automaton detects a symbolic model
wth cycles"? Please clarify.

l.248: You put I'(l_0')=\emptyset, but that's supposed to be a clock
constraint. Do you mean "true"?

l.268f: Same here, both for I'(l_0') and for the guard on the second
added edge.

l.276: "... accelerated automat*a* ..."

l.293f: I didn't understand the sentence starting "The model". Please
reformulate.

l.376: Please remove the first "l_reset".

l.376f: Please clarify your statement by saying "There is a single
acceleratable cycle whose effect is ...".

l.383: The condition "a_j+a_k\ge \forall(a_j, a_k)" is mathematically
meaningless. Please reformulate.

l.404: Please remove the first "l_reset".

l.415: Please remove the first "l_reset".

l.421f: Again some strange occurrences of \emptyset. You probably
mean I'(l_reset')=true, but on the edges in l.422 you twice put a
*label* \emptyset. Do you mean \epsilon here? (There is one more
\emptyset which should be "true", too.)

l.472: "..., from *the* timed ... from *the* accelerated"

l.503: What do you mean "overcome the parallel composition"? Please
clarify.

l.566: What do you mean "the number and scale of access number"?
Please clarify.

Reviewer 2 ·

Basic reporting

#1 Basic Reporting
As I have already this paper, and my previous comments about the
general structure of the paper and readability of the paper remains, I will
not discuss too much here.

The paper is still well-structured and remains readable. This said, I
still find Definition 3 hard to read. The paragraph
afterwards does however help in explaining better the window of an
acceleratable cycle.

Experimental design

One of my comments from my last review was that the papers made claims
about the memory consumption, that was not visible in the paper. In
the rebuttal they comment on this, and I believe they are correct: the
problem is that software does allocate memory in chunks from the
operating system , and so the atual actual higher memory
consumption would not be visible between the two techniques. As the
authors are aware of this, I think it would have been adequate to
mention how memory was measured - and that accelerating cycles does
not (necessarily) reduce memory consumption (it could even increase it
based on the rebuttals mentioning that the state space will increase
due to adding extra locations?).
It does indeed reduce time (no doubt about it), but I keep pushing on
the memory because the authors in the introduction (page 7 line 83)
say Exact Acceleration can reduce required storage space. This is
not clear in the experiments, and they contradict this statement in the
rebuttal. Admittedly, Uppaal (and other model checkers) will usually
not generate the full state space during a search, and so the state
space actually searched with acceleration might be smaller.

Validity of the findings

The paper has made up for the lacking definition of continuous zones -
basically it means that there is an overlap of the visited clock zones in a
given location after repeating the cycle.

Now, In my previous reviews
I said I was unable to validate the the correctness of the paper, as
it hinges on the correctness of Theorem 2 (now Theorem 3). I am
pleased to see, that the authors have updated this proof - but my
objection remains: they set out to prove there exists an integer $n$
such that the clock zones becomes continous afte $n$ iterations of the
cycles. In the proof they start out by stating: "According to the
forward reachability analysis, the problem of fragments in the
acceleratable cycle will inevitably lead to the overlap of clock
zones, that is the appearance of continuous clock zone. So the point
of the proof is to determine the value of the positive integer, that is, when
there is a continuous clock zone" and later "At the reset location, we
assume that the clock zone is continuous from the (n+1)th time onward... essentially stating that their
Theorem is correct, and they just have to determine the value of $n$ (hence my comment about assuming the correctness of their own
Theorem).

Additional comments

As you can see, I'm still not convinced about the proof of Theorem 3. I find the result plausible, but would still recommend a revision of the proof.

---

## Round 0.3 · accepted · Accept

The reviewers and myself are satisfied with the way in which you have addressed their concerns.